# Improvements in Quality of Life and Thyroid Parameters in Hypothyroid Patients on Ethanol-Free Formula of Liquid Levothyroxine Therapy in Comparison to Tablet LT4 Form: An Observational Study

**DOI:** 10.3390/jcm10225233

**Published:** 2021-11-10

**Authors:** Katarzyna Bornikowska, Małgorzata Gietka-Czernel, Dorota Raczkiewicz, Piotr Glinicki, Wojciech Zgliczyński

**Affiliations:** 1Department of Endocrinology, Centre of Postgraduate Medical Education, 01-813 Warsaw, Poland; malgorzata.gietka-czernel@bielanski.med.pl (M.G.-C.); piotr.glinicki@bielanski.med.pl (P.G.); wojciech.zgliczynski@bielanski.med.pl (W.Z.); 2Department of Medical Statistics, School of Public Health, Centre of Postgraduate Medical Education, 01-813 Warsaw, Poland; dorota.bartosinska@gmail.com

**Keywords:** liquid levothyroxine, hypothyroidism, quality of life, questionnaire ThyPROpl, sex hormone binding globulin

## Abstract

Levothyroxine (LT4) is a standard therapy in hypothyroidism; however, its bioavailability and therapeutic effects might be affected by many factors. Data shows that therapy with liquid LT4 characterized by quicker pharmacokinetics provides better thyroid hormones control than tablet LT4. We addressed the quality of life (QoL) and efficacy of the new ethanol-free formula of liquid LT4 (Tirosint^®^SOL) treatment in 76 euthyroid patients with primary (PH, *n* = 46) and central hypothyroidism (CH, *n* = 30), and compared the results to retrospective data on equivalent doses of tablet L-T4 therapy. After 8 weeks of liquid LT4 therapy, we found a significant improvement in QoL in both PH and CH patients. TSH levels were unaltered in PH patients. Free hormone levels (fT4 and fT3) increased in all the patients, with the exception of fT3 in the CH group. SHBG and low-density lipoprotein (LDL) also improved. Liquid LT4 therapy provided a better thyroid hormone profile and improvement in patients’ QoL than the tablet form, which was possibly due to the more favorable pharmacokinetics profile resulting in better absorption, as suggested by the increased free thyroid hormone levels. In summary, this is the first study addressing the QoL in hypothyroid patients, including primary and central hypothyroidism, treated with liquid LT4 formula in everyday practice.

## 1. Introduction

Hypothyroidism is one of the most common diseases of the endocrine system; subclinical hypothyroidism affects up to 3–15% of the adult population worldwide [1,2]. Synthetic levothyroxine (T4) sodium is the first-line treatment for hypothyroidism, regardless of its cause, in all age groups. The goal of treatment with levothyroxine is to achieve clinical and biochemical euthyroidism, evidenced by the achievement of the desired TSH and/or fT4 concentration, while avoiding drug overdose [3,4,5]. It is worth emphasizing that the quality of life of patients treated for primary hypothyroidism is reduced despite biochemical euthyroidism in about 10% of patients [6,7,8]. On average, 70% of orally ingested LT4 is absorbed in the small intestine (duodenum, jejunum, and ileum) and an acidic intragastric pH is required for optimal dissolution of the LT4 tablet [9]. The American and European Thyroid Associations recommend that L-thyroxine be taken with water consistently 60 min before breakfast or at bedtime, 2–4 h after the last meal, for optimal constant absorption [4,10]. Despite this, many different factors, both physiological (pregnancy, elderly age) and pathophysiological (Helicobacter pylori-associated gastritis, autoimmune gastritis, lactose intolerance, celiac disease, small intestinal bacterial overgrowth, and bariatric surgery), and even some food ingredients might influence levothyroxine’s bioavailability and thus its therapeutic effect [11,12,13,14].

Clinical data showed that therapy with liquid LT4 in 85% glycerol and 96% ethanol solutions was more efficient and provided better thyroid hormones control and enabled less frequent need of TSH level monitoring in replacement therapy in hypothyroid patients and TSH suppression therapy, while being an adjunct treatment in well-differentiated thyroid cancer patients [15,16,17,18,19,20,21,22,23,24]. Furthermore, it enabled the elimination of LT4 malabsorption issues related to gastrointestinal diseases, such as gastritis, celiac disease, small intestinal bacterial overgrowth, as well as in patients after bariatric surgery, or who underwent polytherapy [13,15,16,17,25,26,27,28,29,30,31].

The aim of this observational pilot study is to address the efficacy of the new ethanol-free formula of liquid LT4 (Tirosint^®^SOL) in patients with primary (PH) and central hypothyroidism (CH) and to assess the patients’ quality of life (QoL) in everyday clinical practice, compared to previous therapy with tablet form at the same dosage. To our knowledge, this is the first study addressing a new ethanol-free formula of liquid levothyroxine efficacy in real world setting.

We measured the following: (i) the thyroid function parameters, such as serum thyrotropin (TSH), free thyroxine (fT4), free triiodothyronine (fT3) levels (as a primary endpoint); (ii) the QoL with ThyPRO questionnaire (as a secondary endpoint); and (iii) the analyzed SHGB (sex hormone binding globulin) and lipid profiles, reflecting thyroid hormones effects at the tissue level. 

## 2. Materials and Methods

### 2.1. Study Design

This is an investigator-initiated, open-label, uncontrolled, observational, single-site study performed on outpatients on liquid LT4 replacement therapy.

A total of 80 adult outpatients with primary (*n* = 50) or central (*n* = 30) hypothyroidism on liquid LT4 formulation therapy prescribed due to medical recommendations or adherence difficulties were included. Levothyroxine intake was measured according to the SmPC, on an empty stomach, at least 30 min before breakfast, in both the tablet and the liquid form. The liquid form of levothyroxine (Tirosint^®^SOL, IBSA, Lugano, Switzerland), approved by the FDA in 2017, is a LT4 solution in 85% glycerol in monodoses. It was the patients’ decision whether they took the liquid LT4 directly (the option taken by all but one participant) or dissolved in water (the option chosen by one patient).

The inclusion criteria were as follows: age between 18 and 65 years old; primary or central hypothyroidism treated with LT4 for at least 3 months prior to inclusion in the study; TSH values at last control (within 1 month) between 0.38 and 4 uIU/mL; taking a stable dosage of LT4 in tablet form; treatment with the liquid form of LT4 at the point of inclusion in the study at the same dosage, but not for longer than two weeks; the ability to provide meaningful informed consent and to complete the ThyPROpl questionnaire.

The exclusion criteria were as follows: malabsorption of LT4, due to the presence of atrophic gastritis, gastritis associated with *H. pylori* infection, intestinal malabsorption caused by lactose intolerance, celiac disease, previous bariatric surgery, or gastric or intestinal surgery; concomitant therapy with oral contraception, amiodarone, beta-blockers, lithium, orlistat, raloxifene, cholestyramine, interferons, or antacids; allergy or intolerance to the studied drugs, pregnancy, severe hepatic disorders and dysfunction, chronic kidney disease (Modification Of Diet In Renal Disease—MDRD < 30 mL/min/1.73 mq), congestive heart failure, serious psychiatric disorders, the inability to understand the aim of the study and to be compliant, the inability to provide acceptable consent.

A sample size of 80 was selected for the comparison of serum TSH, fT4, and fT3 levels measured after 8 weeks, following inclusion in the study. These values were compared with the last collected respective measurements. For the CH group, the study required a sample size of 43 patients to achieve a power of 90% and a level of significance of 5% (two-sided), for detecting a mean of the differences of 0.76 (after minus before) in TSH levels, assuming the standard deviation of the differences to be 1.2. For the PH group, the study required a sample size of 25 patients to achieve a power of 90% and a level of significance of 5% (two sided), for detecting a mean of the differences of 2.6 (after minus before) in fT4 level, assuming the standard deviation of the differences to be 3.4. The calculation assumed 30% patients lost to follow-up. The study was not designed to detect differences in QoL; therefore this was evaluated as an exploratory end-point. 

Out of the 80 patients included in the study, 76 completed it: 46 with PH, and 30 with CH. In the PH group, four patients were excluded from the final analysis: two discontinued the therapy, describing the taste of the drug as “being too sweet”, and two patients were non-adherent because they did not take the drug on a regular basis, as prescribed.

### 2.2. Laboratory Measurements

Blood samples for the determination of TSH, fT4, fT3, and SHBG serum concentrations and lipid profiles were collected 8 weeks after liquid levothyroxine treatment. Hormone level measurements were performed using chemiluminescence immunoassays (CLIA) using an automated analyzer for serum concentrations of TSH, fT4, fT3, and SHBG (Advia Centaur XP Immunoassay System; Siemens, Germany). The reference ranges of hormone serum concentration were set as follows: TSH- 0.38–4.0 uIU/mL; fT4- 10,29–21,88 pmol/L; fT3- 1.71–3.71 pg/mL. The reference range for SHBG depends on age and gender. Anti-thyroid peroxidase antibodies (TPO-Ab) and antithyroglobulin antibodies (Tg-Ab) were determined in all the samples by using the CLIA method (Advia Centaur XP Immunoassay System; Siemens, Germany); antibody titers were positive above 60 IU/mL. The lipid profiles, including the levels of total cholesterol, HDL-C, LDL-C and triglycerides, were measured with a Cobas 6000 biochemistry analyzer (Roche Diagnostic, Basel, Switzerland).

### 2.3. Quality of Life (QoL)

A ThyPROpl questionnaire was used to measure patients’ quality of life (QoL). ThyPROpl is a translated and linguistically validated (by Sawicka-Gutaj et al.) version of the original ThyPRO questionnaire [18]. ThyPRO is a thyroid-specific quality-of-life questionnaire applicable to patients with benign thyroid disorders, with substantial evidence for its clinical validity and reliability. ThyPRO is composed of 85 questions, summarized in 12 domains, measuring aspects of QoL relevant to thyroid patients. Patients rate their responses for each item on a five-point Likert scale: 0—not at all; 1—a little; 2—some; 3—quite a bit; 4—very much. If >50% of the items in a scale obtain a valid response, then the scale scores are derived by taking the average item scores of each domain and transforming them linearly into a 0–100 scale (item mean * 25), with increasing scores indicating decreasing QoL (i.e., more symptoms or greater impact of disease). The patients completed the ThyPROpl questionnaire (alone or with support of investigator if needed) on the point of their decision to participate in the study, when they retrospectively evaluated the treatment with the tablet form of L-T4 therapy; and, 8 weeks after their inclusion in the study, filled out the questionnaire again, this time evaluating the treatment with the liquid L-T4 form [32]. 

### 2.4. Statistical Analysis

The statistical analysis was carried out using Statistica 13.1 software (STATSOFT, Tulsa, OK, USA).

The categorical variables are presented as *n* (% frequency of group), and the continuous variables as mean with standard deviation (M ± SD) or median with interquartile range (25–75%), depending on normality of data distribution. The data normality was verified using Shapiro–Wilk’s test and visual assessment of the histograms. The mean or median difference in the continuous variables between the two paired measurements was tested using Student’s paired *t*-test or Wilcoxon’s signed rank test, depending on whether the distribution was normal, respectively.

Missing data (e.g., lost to follow-up) were omitted from the analysis.

The significance level was set at 0.05 in all the statistical tests.

## 3. Results

### 3.1. Patients’ Characteristics

All the patients were aged between 18 and 65 years old, 42.5 years on average in PH group and 41.0 years on average in CH group (Table 1). Both groups were predominantly female (96% and 70%, respectively). The BMI (mean ± SD) in the PH and CH groups was initially 25.65 ± 5.80 and 28.07 ± 6.64 kg/m^2^, respectively, and remained stable over the course of the study (25.62 ± 5.81 kg/m^2^, *p* = 0.176 and 28.02 ± 6.61 kg/m^2^, *p* = 0.06 respectively). The average daily dose of LT4 was 82 µg in the PH group and 95 µg in the CH group.

The etiology of hypothyroidism was predominantly autoimmune thyroiditis in the PH group (90% of patients) and pituitary multihormonal insufficiency after tumor resection in the CH group (80%). 

### 3.2. Comparison of TSH, fT4 and fT3 Serum Concentration in Patients on Liquid and Tablet LT4 Therapy

The TSH levels were not significantly altered in the PH patients after 8 weeks of therapy with liquid LT4 in comparison to the tablet L-T4 (1.71 vs. 1.64 mlU/L, *p* = 0.773), and reduced in the CH patients from 0.25 to 0.16 mlU/L (*p* = 0.037), (Figure 1).

Liquid LT4 treatment resulted in a significant increase in free hormone concentrations: fT4 increased from 14.13 to 15.96 pmol/L (*p* < 0.001) and from 14.15 to 15.30 pmol/L (*p* = 0.003) in the PH and CH groups, respectively (Figure 2).

Moreover, fT3 concentration in the PH patients increased from 2.82 pg/mL when measured during therapy with LT4 in tablet form to 3.15 pg/mL after 8 weeks of therapy with liquid LT4 (*p* < 0.001). In the CH patients, fT3 concentration remained stable over time (2.69 vs. 2.74 pg/mL on average; *p* = 0.447) (Figure 3).

### 3.3. Assessment of SHBG Serum Concentration 

Liquid LT4 therapy led to an increase of SHBG concentration when compared to previous therapy with tablet LT4 form: in the PH patients, this increase was from 54.2 to 56.4 nmol/L on average (*p* = 0.033), and in the CH group, it was from 39.05 to 43.45 nmol/L on average (*p* = 0.026) (Figure 4).

### 3.4. Comparison of LDL-C Serum Concentration 

LDL levels significantly decreased after 8 weeks of therapy with liquid LT4 in comparison to tablet LT4 in the PH patients (from 97.91 to 88.17 mg/dL on average, *p* = 0.049); however, LDL levels remained unaltered in the CH group (107.54 vs. 103.49 mg/dL on average, *p* = 0.259), (Figure 5). Total cholesterol, HDL-C, and triglyceride levels were not significantly changed at the same time (*p* > 0.05). 

### 3.5. Changes in Quality of Life in Patients on Liquid and Tablet L-T4 Therapy

We examined QoL in patients using the ThyPROpl questionnaire at two timepoints: retrospectively, at the beginning of the study, when the patients judged their QoL when o taking LT4 in tablet form, and for the second time after 8 weeks of therapy with liquid levothyroxine. We found a significant improvement in overall quality of life on liquid LT4 therapy in patients irrespective of their hypothyroidism’s etiology, expressed through a QoL score reduction: the average score in the PH patients was 37.5 vs. 24.5, *p* = 0.001; in the CH group, it was 42.5 vs. 17.5 (*p* < 0.001). We noticed significantly improved outcomes in all of 12 domains in the PH patients and in 10 of 12 domains in the CH patients (Table 2 and Appendix A).

## 4. Discussion

The present observational study showed for the first time that the novel, ethanol-free liquid LT4 is more effective than tablet LT4 in increasing the concentration of free thyroid hormones in euthyroid patient populations with primary and central hypothyroidism. Moreover, we addressed the effect of TSH levels on liquid LT4 therapy and, as with the fT3 and fT4 measurements, we compared them with retrospective values that were measured when the same patients were taking LT4 in tablet form.

Most previous studies assessed the liquid thyroxine solution containing 85% glycerol and 96% ethanol [15,16,17]. 

An observational, prospective study on 141 hypothyroid patients without LT4 malabsorption by Fallahi et al. [18] showed that the ethanol solution formula of liquid LT4 was more effective than the tablet form of LT4 at controlling TSH levels. TSH values significantly declined with respect to the basal value after switching to liquid LT4 both at the first follow-up at 1–3 months (*p* < 0.05) and at the second at 5–7 months (*p* < 0.01). FT4 and FT3 levels remained unaltered [18]. In another study [19], the authors obtained similar results from patients without malabsorption after total thyroidectomy due to differentiated thyroid cancer. The data showed that the use of the LT4 liquid formulation, as compared to tablets, resulted in a significantly higher number of patients maintaining TSH values within the target range recommended by the American Thyroid Association. Furthermore, the variability in TSH concentrations in this patient population was reduced over time [20]. In the present study, TSH serum concentrations were unaltered in the PH patients on liquid LT4 therapy, which contrasts with the results presented in the aforementioned studies [18,19]. In our opinion, TSH serum concentration was not significantly altered in the PH group for several potential reasons: the short follow-up, the small number of patients (*n* = 30), the individual efficiency of enzymatic deiodase II, and the central availability of triiodothyronine, which inhibits the transcription of genes encoding both TSH subunits. Moreover, recent large population studies have shown that the best fitted model of the TSH–fT4 relationship is complex and depends on individual hypothalamic-pituitary-thyroid axis equilibrium point, sex, age, and thyroid function status [33,34,35,36]. The standard model of thyroid homeostasis postulates a logarithmic relationship between FT4 levels and pituitary TSH release. Hadlow et al. found that in LT4-treated hypothyroid patients, the TSH response to fT4 changes is attenuated (as evidenced by a lesser slope in the TSH-fT4 curves) when fT4 serum concentration is within the reference range, and especially within the range from 15 to 20 pmol/L [33,34]. Thus, it is difficult to estimate the minimal required change in serum fT4 concentration that would effectively alter serum TSH levels. We hypothesize that, in the present study, the eight-week time of exposure of the PH patients to a significant increase in fT4 serum concentration, induced by liquid LT4 therapy (on average by 2 pmol/L, from 14.13 to 15.96 pmol/L, *p* < 0.001), was not long enough to alter TSH secretion significantly in euthyroid patients.

Interestingly, we found a simultaneous and significant increase in fT4 and fT3 concentrations in the CH patients on liquid LT4 therapy. Our results in the group of 30 patients with central hypothyroidism are in agreement with those of a prospective study by Benvenga et al., which evaluated 13 patients with isolated central hypothyroidism. After switching to liquid LT4, serum fT4 concentrations significantly increased. This was most likely related to the more favorable pharmacokinetic profile of liquid levothyroxine compared with the tablet form [37]. This might indicate the liquid LT4’s superior absorption to the tablet LT4 formulation, which was also suggested by the significantly decreased TSH levels in the CH group in our study. Although the measurement of TSH concentration in patients with central hypothyroidism offers only limited value for the assessment of thyroid function, it may provide additional information as to the efficiency of LT4 replacement therapy depending on the etiology of CH (congenital or acquired) [38]. In acquired central hypothyroidism (in our study, this applied to 28/30 patients in the CH group), TSH is usually detected within the normal range (14/28 in our CH group) because of well-preserved immunoreactivity; however, the disease is characterized by a severe impairment of intrinsic bioactivity and the ability to stimulate TSH receptors. Ferretti et al. reported that about half of the target LT4 substitutive dose is sufficient to suppress TSH secretion in about 80% of CH patients, even though serum FT4 levels are still in the hypothyroid range in most cases [38]. Shimon et al. suggested that in central hypothyroidism, baseline TSH is usually within normal range, and is further suppressed by exogenous thyroid hormone, as in primary hypothyroidism, but to a lesser extent. Thus, insufficient replacement may be reflected by inappropriately elevated TSH levels, and may lead to dosage increment [39]. 

Further, we investigated the impact of levothyroxine treatment on the tissue level by measuring SHBG and LDL-C concentrations. The SHBG serum concentrations increased and the LDL-C decreased after liquid LT4 therapy, suggesting the higher effectiveness of liquid than tablet LT4. To the best of our knowledge, changes in SHBG concentration and lipid profile after switching from tablet LT4 to liquid formulation were not assessed in previous studies. Thyroid hormones, especially T3, stimulate SHBG liver synthesis and improve lipid profiles [40,41,42]. Along with osteocalcin and angiotensin convertase, SHBG and LDL-C levels, are alternative biomarkers through which to assess the supply of thyroid hormones in an organism [3,43,44,45]. It is worth mentioning that reduced SHBG levels are associated with insulin resistance and obesity due to hyperinsulinemia, which decreases the liver synthesis of SHBG [46]. Since, in our study group, body mass index remained stable over the follow-up and all of the obese patients (9/46 in PH group and 8/30 in CH group) were treated with the same dose of metformin during the observation period and the previous six months, we ascertained that the observed increase in SHBG concentration was not affected by changes in insulinemia, but instead caused by thyroid hormones. In our opinion, an increase in SHBG and a decrease in LDL-C levels indicates the better peripheral effect of thyroid hormones on liquid LT4 therapy. 

These results are in line with those of previous studies showing that liquid LT4 is more effective than the tablet form in patients without [15,16,17] and with LT4 malabsorption caused by gastrointestinal disease, medications, and bariatric surgery, and in special groups of patients with hypothyroidism, such as: the elderly, children, and pregnant women [23,24,25,26,27,28,29,30,31,46,47,48,49]. Nevertheless, the assessment of gene products whose transcription depends on the concentration of thyroid hormones could be a more reliable measure in terms of the peripheral effects of thyroid hormones than SHBG and LDL-C concentration, and is worth of exploring in future research.

Finally, the disturbance of the function of the thyroid has a negative impact on patients’ QoL [32]. Even when biochemical euthyroidism is achieved, QoL might still be reduced and suboptimal for patients’ comfort [6,7,8]. Differences in fT4 and TSH concentrations, even within the reference range, may be a determinant of psychological well-being in treated hypothyroid patients [50]. The outcome of hormonal therapy should not be judged exclusively using the laboratory measurements, as QoL determines patients’ wellbeing, impacting their overall life and their satisfaction with the therapy [50]. In the present study, the therapy with liquid levothyroxine yielded substantially improved QoL in both the investigated patient groups; in particular, the patients reported reduced symptoms of hypothyroidism and less tiredness and nervousness, and claimed to experience fewer difficulties with daily activities. Due to the observational character of the study, we could not mask the drug formula. We also cannot exclude the effect of the novelty on the patients; however, in such a case, it can be assumed that, depending on the patients’ attitude (positive or negative towards novel treatments), both the improvement and the worsening of subjective wellbeing will be observed. However, improvements were observed in 12/12 of the investigated domains in the PH and in 10/12 in the CH group. Improvement in QoL is a crucial outcome of successful therapy, and given that hypothyroidism therapy is a lifelong intervention, it has to be not only effective in terms of biochemical equilibrium, but also, most importantly, in terms of satisfying patients’ well-being and convenience.

## 5. Conclusions

Our observational study levothyroxine therapy for patients with primary and central hypothyroidism revealed that liquid LT4 displays higher efficacy, providing a better thyroid hormone profile and a greater improvement in patients’ QoL than the tablet LT4 form. This might be due to a more favorable pharmacokinetics profile, resulting in better absorption of the liquid over the tablet LT4 form, as revealed by the increased free thyroxine levels we observed. In summary, this is the first study addressing the QoL of hypothyroid patients, including primary and central hypothyroidism, further confirming the efficacy of liquid levothyroxine.

## Figures and Tables

**Figure 1 jcm-10-05233-f001:**
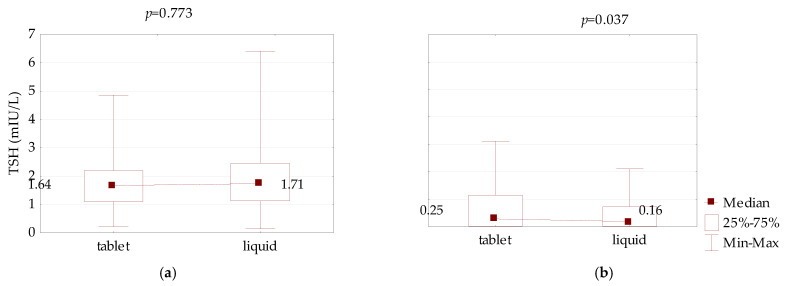
Comparison of TSH serum concentration between LT4 therapy in tablet and liquid form in an equivalent dose in: (**a**) primary hypothyroidism and (**b**) central hypothyroidism.

**Figure 2 jcm-10-05233-f002:**
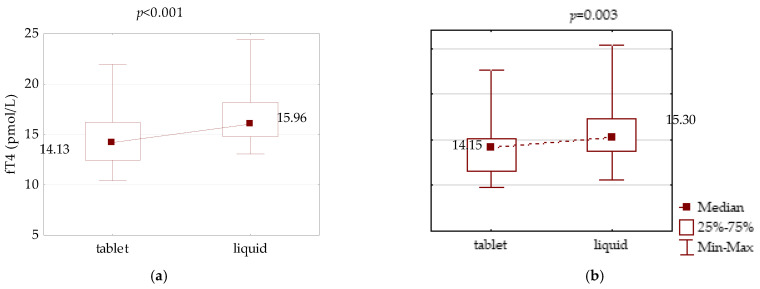
Comparison of fT4 serum concentration between LT4 therapy in tablet and liquid form in an equivalent dose in: (**a**) primary hypothyroidism and (**b**) central hypothyroidism.

**Figure 3 jcm-10-05233-f003:**
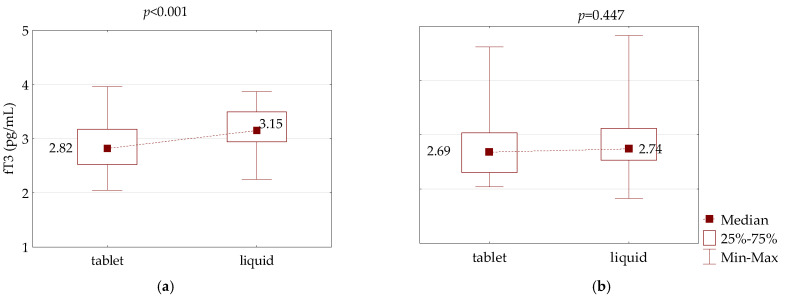
Comparison of fT3 serum concentration between LT4 therapy in tablet and liquid form in an equivalent dose in: (**a**) primary and (**b**) central hypothyroidism.

**Figure 4 jcm-10-05233-f004:**
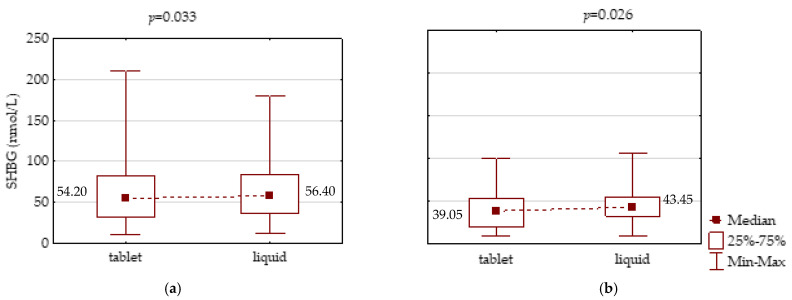
Comparison of SHBG serum concentration between LT4 therapy in tablet and liquid form in an equivalent dose in: (**a**) primary and (**b**) central hypothyroidism.

**Figure 5 jcm-10-05233-f005:**
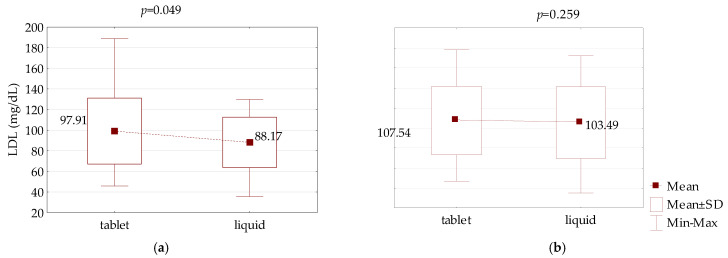
Comparison of LDL-C serum concentration between LT4 therapy in tablet and liquid form in an equivalent dose in: (**a**) primary hypothyroidism and (**b**) central hypothyroidism.

**Table 1 jcm-10-05233-t001:** Clinical characteristics of patients with primary and central hypothyroidism.

Variable (Unit), Parameter	Primary Hypothyroidism (*n* = 46)	Central Hypothyroidism (*n* = 30)
Age (years), M ± SD	42.5 ± 14.2	41.0 ± 14.2
Gender, female, *n* (%)	44 (95.65)	21 (70.00)
male, *n* (%)	2(4.35)	9 (30.00)
BMI (kg/m^2^), M ± SD	25.65 ± 5.80	28.07 ± 6.64
L-T4 daily dose (µg), M ± SD	82 ± 33	95 ± 47
Etiology of hypothyroidism, *n* (%)	Autoimmune thyroiditis, 41 (90.00)	Pituitary insufficiency after tumor resection, 24 (80.00)
	Postsurgical hypothyroidism 5 (10.00)	Other, 6 (20.00)

M—mean, SD—standard deviation, BMI—body mass index.

**Table 2 jcm-10-05233-t002:** Comparison of QoL (ThyPROpl) between LT4 therapy in tablet and liquid form in an equivalent dose in primary and central hypothyroidism.

Domain	Primary Hypothyroidism (*n* = 46)			Central Hypothyroidism (*n* = 30)		
	Tablet	Liquid	*p*	Tablet	Liquid	*p*
Hypothyroid symptoms	37.5 ± 21.5	27.2 ± 16.4	<0.001	29.4 ± 20.5	16.9 ± 19.4	<0.001
Tiredness	56.0 ± 26.6	35.5 ± 23.9	<0.001	47.3 ± 31.4	26.5 ± 24.2	0.005
Lack of vitality	65.2 ± 21.6	56.3 ± 23.0	0.033	57.8 ± 25.1	48.8 ± 23.4	0.107
Memory and concentration	34.4 ± 27.9	20.6 ± 18.7	<0.001	25.6 ± 20.2	19.5 ± 19.3	0.140
Nervousness	38.0 ± 27.4	22.4 ± 19.8	<0.001	30.6 ± 20.3	16.8 ± 14.0	0.003
Psychological well-being	42.7 ± 25.0	34.5 ± 22.1	0.011	40.1 ± 27.8	27.7 ± 21.6	0.024
Mood swings	36.3 ± 26.6	28.7 ± 21.9	0.010	34.3 ± 20.3	21.9 ± 17.6	0.013
Relationships	18.4 ± 23.1	13.6 ± 20.0	0.047	24.4 ± 28.7	8.9 ± 10.1	0.003
Daily activities	22.9 ± 27.7	13.4 ± 19.6	<0.001	25.2 ± 24.7	12.5 ± 13.4	0.018
Sex life	39.8 ± 38.3	31.8 ± 31.7	0.048	32.1 ± 33.7	15.8 ± 26.7	0.004
Appearance	26.5 ± 25.2	19.7 ± 23.5	0.014	29.4 ± 31.1	16.9 ± 20.7	<0.001
Overall	37.5 ± 31.1	24.5 ± 26.6	0.001	42.5 ± 30.9	17.5 ± 19.9	<0.001

Results are presented as M ± SD, M—mean, SD—standard deviation, *p* for Student’s paired *t*-test.

## Data Availability

The data presented in this study are available on request in the Department of Endocrinology, Centre of Postgraduate Medical Education, 01-813 Warsaw, Poland. The data are not publicly available due to privacy restrictions.

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
