# Peer review of "Improvements in Quality of Life and Thyroid Parameters in Hypothyroid Patients on Ethanol-Free Formula of Liquid Levothyroxine Therapy in Comparison to Tablet LT4 Form: An Observational Study"

_jcm, 2021, doi:10.3390/jcm10225233_

Round 1

Reviewer 1 Report

The authors of this manuscript have compared the effects of a novel liquid preparation of T4 in hypothyroid patients previously treated with tablet thyroxine.

The main concern of this Referee comes from some key methodologic approaches:

There is no mention on how were treated patients while on tablet T4. Do they waited half an hour or one hour following pill ingestion before breakfast? Same question could be made about the liquid preparation: it was a droplet preparation or in monodose , it was taken pure or in a glass of water?

The dose of tablet T4 is reported without individually tailoring and this may represent a significant bias. In fact, looking at the table 1 it can be argued that some of your patients were obese which implies different doses of T4 and different meaning of serum TSH

SHBG is not a good peripheral marker in obese patients in which hyperinsulinemia may lower the protein as compared with the non obese

There may be a significant difference of pharmacologic thyroid homeostasis in hypothyroid patients showing a serum TSH around 0.38 or around 4 μU/ml. Better individualization of sample would improve the data

Intra and Interassay variations for TSH and thyroid hormones assay  should be also mentioned in view of the narrow differences observed in the results section

QoL evaluation requires a follow up longer than 8 wks to avoid important bias coming from the patient's mind, such as the novelty of treatment and the hope of improving. In particular, it seems a bit surprising that, in euthyroid patients, the hypothyroid symptoms and the related psichologic (and not measurable) feelings might have improved so quickly.

Minor Points

line 99: perhaps is CH and not PH

A number of outdated references between n.22 and n. 37 may be removed because already analyzed in a meta-analysis (Levothyroxine Therapy: Changes of TSH Levels by Switching Patients from Tablet to Liquid Formulation. A Systematic Review and Meta-Analysis. Front Endocrinol (Lausanne). 2018 Jan 26;9:10 and in an updated review focused on this topic (Novel thyroxine formulations: a further step toward precision medicine. Endocrine. 2019 Oct;66(1):87-94.

Author Response

Comments and Suggestions for Authors

The authors of this manuscript have compared the effects of a novel liquid preparation of T4 in hypothyroid patients previously treated with tablet thyroxine.

The main concern of this Referee comes from some key methodologic approaches:

REMARK 1: There is no mention on how were treated patients while on tablet T4. Do they waited half an hour or one hour following pill ingestion before breakfast? Same question could be made about the liquid preparation: it was a droplet preparation or in monodose , it was taken pure or in a glass of water?

ANSWER 1: Thank you for this comment, in the present form of the manuscript we described the treatment schedule (which is in line with Tirosint®SOL SmPC) more into detail in chapter 2.1. Study design, please see page 2 lines 78-83:

Levothyroxine intake was performed according to the SmPC, on empty stomach, at least 30 min before breakfast in case of both tablet and liquid form. Liquid form of levothyroxine (Tirosint®SOL, IBSA, Switzerland), approved by FDA in 2017, is a LT4 solution in 85% glycerol in monodoses. It was patients’ decision if they take liquid LT4 directly (everyone with exception of one person) or dissolved with some water (one patient).

REMARK 2: The dose of tablet T4 is reported without individually tailoring and this may represent a significant bias. In fact, looking at the table 1 it can be argued that some of your patients were obese which implies different doses of T4 and different meaning of serum TSH.

ANSWER 2: The doses of tablet LT4, which were the same as the dose of liquid LT4 during the study, were adjusted to the individual patient needs accordingly to our best clinical practice and experience. We would like to draw the attention of Reviewer 1, that all patients included, independently from hypothyroidism etiology, were in euthyroid state before the study started, as we stated in the previous and present version of the manuscript, please see page 2, lines 84-89:

Inclusion criteria were as follows: age between 18 and 65 years old, primary or central hypothyroidism treated with LT4 for at least 3 months prior inclusion into the study, TSH values at last control (within 1 month) between 0,38 and 4 uIU/mL, under stable dosage of LT4 in tablet, treatment with liquid form of LT4 at the moment of inclusion into the study at the same dosage, but not longer than 2 weeks, ability to give meaningful informed consent and to complete the ThyPROpl questionnaire.“

Moreover, the BMI remained stable over the course of the study- at the time to inclusion to our observational study BMI was 25,65 kg/m2 on average in PH group and 28,07 kg/m2 on average in CH group. BMI was 25,62 kg/m2 and 28,02 kg/m2 on average respectively after eight weeks treatment of liquid LT4. Furthermore, the trial was not designed in order to stratify the patients into subgroups according to their BMI.

Taking into account the comment on BMI by Reviewer 1, in the present form of the manuscript we comment about stable BMI over the course of the study in chapter 3.1 Patient characteristics, please see page 4 lines 162-164:

,,BMI (mean±SD) in PH and CH group was initially 25.65±5.80 and 28.07±6.64 kg/m2 respectively and remained stable over the course of the study (in PH group 25.62±5,81kg/m2, p=0.176 and 28.02± 6,61 kg/m2, p=0.06, respectively)”

Furthermore, the trial was not designed in order to stratify the patients according to BMI.

REMARK 3: SHBG is not a good peripheral marker in obese patients in which hyperinsulinemia may lower the protein as compared with the non obese

ANSWER 3: Thank you for this comment. We use the marker SHGB as “alternative” and with caution which we stated in the previous (and present) version of the manuscript (chapter: Discussion):

SHBG and LDL-C levels, next to osteocalcin and angiotensin convertase, are alternative biomarkers to assess the supply of the organism with thyroid hormones [3, 47,48].” page 9 lines 299-301

the assessment of gene products whose transcription depends on the concentration of thyroid hormones could be more reliable measure in terms of peripheral effects of thyroid hormones than SHBG and LDL-C concentration and is worth of exploring in the future research.”

page 9 lines 317-320

Moreover, our choice of SHBG as an additional measure we based on suggestions by the authors Jonklaas et al (2014) in “Guidelines for the Treatment of Hypothyroidism: Prepared by the American Thyroid Association” (ref. 3) where they claimed that SHBG can be used as an additional marker, especially in situation, when TSH concentration levels are within the normal range: ,,End-organ markers of response to thyroid hormone include sex hormone binding globulin (SHBG), osteocalcin, urinary n-telopeptides, total cholesterol, low-density lipoprotein (LDL) cholesterol, lipoprotein(a), creatine kinase, ferritin, myoglobin, and enzymes such as tissue plasminogen activator, angiotensin converting enzyme (ACE), and glucose 6-phosphate dehydrogenase (61,75–81). The results of RCTs have indicated that cholesterol (50,82–86), and SHBG levels (50) are particularly affected by the administration of LT4. Similarly, RCTs have demonstrated that LT4 replacement therapy affects myocardial function (87) and particularly diastolic function (88–92), and, over time, the brachial artery intimal thickness (89,93). Of note, the changes observed in these tissue markers of thyroid hormone action are often within the range of variance of the normal population, so these tests may be considered as an additional tool to allow a further optimization of the replacement therapy, once the TSH is already within target range”

Furthermore, we would like to draw the attention of Reviewer 1 that we used SHGB and lipid profiles measurements only as additional parameters measured during the study.

Regarding the Reviewers 1 point about hyperinsulinemia – indeed, obesity and particularly excess visceral fat are associated with decreased SHBG levels in both sexes due to hyperinsulinemia which decreases liver production of SHBG. However, in our study patients with BMI >= 30 kg/m2 (9/46 in PH group and 8/30 in CH group) were treated with metformin for several years and its dose remained unchanged during the observation period and the past six moths. Also, we observed a significant increase of SHBG in both groups after eight weeks of therapy with liquid LT4 while BMI was stable over the course of the study: at the time to inclusion to our observational study BMI was 25,65 kg/m2 on average in PH group and 28,07 kg/m2 on average in CH group and it was stable after eight weeks treatment of liquid LT4- 25,62 kg/m2 and 28,02 kg/m2 on average respectively.

In the present form of the manuscript we improved the fragment describing the issue of SHBG concentration in chapter 4 Discussion, please see page 9 lines 301-309

It is worth mentioning that reduced SHBG levels are associated with insulin resistance and obesity due to hyperinsulinemia which decreases liver synthesis of SHBG [51,52]. As in our study group body mass index remained stable over the follow-up and all of the obese patients (9/46 in PH group and 8/30 in CH group) were treated with the same dose of metformin during the observation period and the past six months we can ascertain that the observed increase in SHBG concentration was not affected by changes in insulinemia but caused by thyroid hormones. In our opinion an increase of SHBG and decrease of LDL-C level indicate a better peripheral effect of thyroid hormones on liquid LT4 therapy”

REMARK 4: There may be a significant difference of pharmacologic thyroid homeostasis in hypothyroid patients showing a serum TSH around 0.38 or around 4 μU/ml. Better individualization of sample would improve the data

ANSWER 4:

This is an interesting suggestion; however in the current study the sample size was not calculated taking into account further stratification of patients. Indeed, in our future research it would be of high interest to include much bigger sample size and perform stratification of the euthyroid patients accordingly to their TSH level to address the question if the impact of altered LT4 formula might be more profound in one of the groups.

REMARK 5: Intra and Interassay variations for TSH and thyroid hormones assay should be also mentioned in view of the narrow differences observed in the results section

ANSWER 5: Thank you for this comment. The intra- and interassay variations are in general an important issue; however in the current study Intra- and interassay variations were at very low level- as we present below and therefore we did not addressed them in the main text of the manuscript:

Intra-assay coefficients of variation (CV) for TSH were: 2,9% for 1,03 μIU/ml, 2,4% for 5,4 μIU/ml, and 1,7% for 10,7 μIU/ml, respectively. Inter-assay CV values were: 2,4% for 1,03 μIU/ml, 0,9% for 5,4 μIU/ml, and 1,2% for 10,7 μIU/ml, respectively.

Intra-assay CV for fT3 were: 2,4% for 1,1 pg/ml, and 2,0% for 3,7 pg/ml, respectively. Inter-assay CV were: 1,5% for 1,1 pg/ml, and 3,7 pg/ml, respectively.

Intra-assay CV for fT4 were: 3,3% for 9,3 pmol/l, 2,2% for 19,0 pmol/l and 2,5% for 38,8 pmol/l, respectively. Inter-assay CV were: 4,2% for 9,3 pmol/l, 4,6% for 19,0 pmol/l, and 3,4% for 38,8 pmol/l, respectively.

REMARK 6: QoL evaluation requires a follow up longer than 8 wks to avoid important bias coming from the patient's mind, such as the novelty of treatment and the hope of improving. In particular, it seems a bit surprising that, in euthyroid patients, the hypothyroid symptoms and the related psichologic (and not measurable) feelings might have improved so quickly.

ANSWER 6: Indeed, we cannot exclude the effect of the novelty in patients which we discuss in the present version of the manuscript in chapter 5 Discussion:

page 9, lines 332-337: “Due to observational character of the study we could not mask the drug formula. We also cannot exclude the effect of the novelty in patients; however in such a case it is imaginable that depending on patients’ attitude (positive or negative towards novel treatments): both: improvement or worsening of subjective wellbeing could be observed. However the improvements were observed in 12/12 of investigated domains in PH and in 10/12 in CH group.”

Indeed, we agree with reviewer 1 that a substantially longer observation time could help to minimize the putative “novelty effect” (whether positive or negative) and in future research it would be reasonable to address QoL over a longer time period.

Minor Points

REMARK 1: line 99: perhaps is CH and not PH

ANSWER 1: Thank you for this comment, yes indeed, it should be CH, we corrected the mistake accordingly.

REMARK 2: A number of outdated references between n.22 and n. 37 may be removed because already analyzed in a meta-analysis (Levothyroxine Therapy: Changes of TSH Levels by Switching Patients from Tablet to Liquid Formulation. A Systematic Review and Meta-Analysis. Front Endocrinol (Lausanne). 2018 Jan 26;9:10 and in an updated review focused on this topic (Novel thyroxine formulations: a further step toward precision medicine. Endocrine. 2019 Oct;66(1):87-94.

ANSWER 2: The style of citing references we have aligned with “Recommendations for the Conduct, Reporting, Editing, and Publication of Scholarly Work in Medical Journals” (updated December 2019) (http://www.icmje.org/icmje-recommendations.pdf) published at the website of ICMJE (International Committee of Medical Journals Editors) claiming that : “Authors should provide direct references to original research sources whenever possible” (page 17, chapter g. References; i. general considerations). Therefore we cited original works in the previous version of the manuscript; however – in line with suggestion of the Reviewer 1 - we cite the above-mentioned meta-analysis and review as valuable sources in the present version of the manuscript in chapter Introduction, lines 57 and in chapter Discussion, lines 237.

Reviewer 2 Report

jcm-1424356 comments: Improvements of quality of life and thyroid parameters in hypothyroid patients on ethanol-free formula of liquid levothyroxine therapy in comparison to tablet LT4 form: an observational study.

This study aims to assess the efficacy of the ethanol-free formula of liquid LT4 in patients with primary (PH) and central hypothyroidism (CH), compared to therapy with tablet form at the same dosage

The article is well-written and would be of interest to the readership of Journal of Clinical Medicine. However, there remain several aspects that require correction or clarification:

  1. The Authors trying to explain reasons of no changes in TSH plasma concentrations in group PH patients (eg. short follow-up), meanwhile, almost thoughtlessly, lower concentrations of TSH in CH group - the same time of follow-up. Please clarify.
  2. Sometimes, during treatment with levothyroxine patients complain of hyperhidrosis, despite the lack of TSH concentration abnormality. Did that symptom change in your observation?

Minor Revisions

References should be described according to JCM Instruction for Authors

Author Response

Comments and Suggestions for Authors

jcm-1424356 comments: Improvements of quality of life and thyroid parameters in hypothyroid patients on ethanol-free formula of liquid levothyroxine therapy in comparison to tablet LT4 form: an observational study.

This study aims to assess the efficacy of the ethanol-free formula of liquid LT4 in patients with primary (PH) and central hypothyroidism (CH), compared to therapy with tablet form at the same dosage

The article is well-written and would be of interest to the readership of Journal of Clinical Medicine. However, there remain several aspects that require correction or clarification:

REMARK 1: The Authors trying to explain reasons of no changes in TSH plasma concentrations in group PH patients (eg. short follow-up), meanwhile, almost thoughtlessly, lower concentrations of TSH in CH group - the same time of follow-up. Please clarify.

ANSWER 1: It is known that TSH concentration measurement in patients diagnosed with central hypothyroidism is not reliable parameter for the assessment of thyroid function. Intriguingly, we observed a significant decrease of serum TSH in this group. We discuss this finding in the present version of the manuscript in chapter Discussion, page 8 lines 275-289:

Circulating TSH level has a limited value for monitoring LT4 treatment in CH patients, but it may provide additional information about LT4 replacement therapy depending on the etiology of CH (congenital or acquired). In acquired central hypothyroidism ( in our study two patients have a congenital CH, others have an acquired CH) baseline TSH is usually within normal range, but it’s not bioactive.

We discuss about it in present version of manuscript in chapter Discussion lines 275-289

...........

Please see the articles (ref. 38):

Luca Persani, Central Hypothyroidism: Pathogenic, Diagnostic, and Therapeutic Challenges, The Journal of Clinical Endocrinology & Metabolism, Volume 97, Issue 9, 1 September 2012, Pages 3068–3078.

,,Defects in TSH secretion may be quantitative (the so-called reduced “TSH reserve”) and/or qualitative (1–4, 9–11). Indeed, serum TSH is low in most cases of genetic CH, a typical example being TSHβ gene mutations that result in the synthesis of a truncated subunit unable to dimerize with the α-GSU (4, 12, 13). In contrast, in acquired CH, the quantitative defect in TSH-producing cells is frequently associated with a qualitative defect in the secreted TSH isoforms that conserve immunoreactivity but display a severe impairment in intrinsic bioactivity and ability to stimulate TSH receptors.’’

,,Several recent papers dealing with substitutive LT4 therapy in patients with CH have underlined the difficulty in achieving optimal replacement. LT4 replacement is easily tuned in primary hypothyroidism by evaluating circulating TSH levels, but this index has a limited value for monitoring LT4 treatment in CH patients (63, 85). However, the finding of unsuppressed serum TSH levels during LT4 treatment strongly indicates undertreatment. Indeed, Ferretti et al. (60) reported that about half of the final LT4 substitutive dose is sufficient to suppress TSH secretion in about 80% of CH patients, despite serum FT4 levels still in the hypothyroid range in most. Similarly, other authors observed that the large majority of 135 CH patients had subnormal serum TSH concentrations during apparently adequate LT4 treatment (86). Subsequently, Shimon et al. (87) suggested that TSH levels above 1.0 mU/liter reflect an insufficient LT4 replacement.”

REMARK 2: Sometimes, during treatment with levothyroxine patients complain of hyperhidrosis, despite the lack of TSH concentration abnormality. Did that symptom change in your observation?

ANSWER 2: None of our patients reported hyperhidrosis during treatment of LT4 in tablet and liquid form.

Minor Revisions

References should be described according to JCM Instruction for Authors

We changed accordingly the references.

Round 2

Reviewer 1 Report

No further suggestions

Reviewer 2 Report

Thank you for your answers. I accept this version of manuscript.